# Zero-shot evaluation of promptable foundation models for 3D CT segmentation

## Abstract

Foundation Models (FMs) have revolutionized interactive segmentation for medical imaging. However, the increasing number of promptable FMs, along with evaluations varying in dataset, metrics, and compared models, makes direct comparison difficult and complicates the selection of the most suitable model for specific clinical tasks. In the context of bone segmentation in CT scans, we evaluated 11 promptable FMs using non-iterative 2D and 3D prompting strategies on both a private and public dataset. The models were categorized based on their prediction dimensionality (2D vs. 3D), and the Pareto-optimal models were identified.

## 1 Introduction

Promptable FMs can serve different purposes, from accelerating annotations (1) to being fine-tuned and adapted for specific datasets (2). Selecting the most suitable model depends on the task at hand, the available resources, and the required accuracy. With the growing number of models, evaluations are typically conducted in isolation, making direct comparison challenging. Although broader benchmarks have been proposed (3; 4) to demonstrate generalization across diverse datasets, clinical applications often demand solutions for specific tasks. In this work, we focus on bone segmentation in CT scans and evaluate promptable FMs, comparing them in the Pareto sense to avoid prioritizing a single metric while also incorporating computational resources.

## 2 Method

### 2.1 Promptable Foundation Models

The introduction of the Segment Anything Model (SAM) (5) made Foundation Models (FMs) popular for interactive segmentation. Building on SAM, SAM2 introduced a memory mechanism allowing video segmentation by means of prediction propagation from initial frames to whole videos. While developed for natural images and videos, initial evaluation studies showed promising results but also evident limitations for medical image segmentation, which motivated the development of dedicated medical FMs. Med-SAM (6), SAM-Med2D (7), ScribblePrompt-SAM (8) and MedicoSAM (9) are SAM-based models fine-tuned for medical data. MedicoSAM (9) also uses prompt propagation for volumetric predictions. SAM-Med3D (10) extends SAM to a 3D architecture trained from scratch. Other 3D interactive segmentation models include SegVol (11), incorporating semantic prompts alongside geometric prompts, Vista3D (12), supporting automatic closed- and interactive open-set segmentation, and nnInteractive (4), offering a well-integrated framework with extended prompting strategies. Med-SAM2 (1) is a SAM2-based model fine-tuned on medical data. All mentioned models have in common that they are promptable with sparse prompts, i.e., the object of interest is identified by a bounding box or point/click, allowing interactive segmentation. Based on the prediction dimensionality (2D vs. 3D) and evaluation manner (slice-wise or volumetric), promptable FMs can be categorized into two groups (Table 1):

Submitted to 39th Conference on Neural Information Processing Systems (NeurIPS 2025). Do not distribute.

Table 1: Overview of promptable FMs with model backbone architecture and the supported prompting strategies. Boxed settings are possible across different models resulting in our default settings. (✓) denotes that authors explicitly stated that the test set of (15) was excluded from training. N denotes that multiple prompts (i.e., up to N prompts) per slice were used.

| Model | Arch. | Med.FT | (15) | Box (1/N) | Point (1/N) | Pt+Box (1/N) | Slice (1/N) | Vol. Limits |
|---|---|---|---|---|---|---|---|---|
| SAM (5) | ViT | ✗ | ✗ | ✓/✓ | ✓/✓ | ✓/✓ | - | - |
| SAM2 2D (13) | Hiera | ✗ | ✗ | ✓/✓ | ✓/✓ | ✓/✓ | - | - |
| Med-SAM (6) | SAM | ✓ | ✓ | ✓/✓ | ✗/✗ | ✗/✗ | - | - |
| SAM-Med2D (7) | SAM | ✓ | ✓ | ✓/✓ | ✓/✓ | ✓/✓ | - | - |
| ScribblePrompt-U (8) | UNet | ✓ | (✓) | ✓/✓ | ✓/✓ | ✓/✓ | - | - |
| ScribblePrompt-SAM (8) | SAM | ✓ | (✓) | ✓/✓ | ✓/✓ | ✓/✗ | - | - |
| MedicoSAM 2D (9) | SAM | ✓ | ✓ | ✓/✓ | ✓/✓ | ✓/✓ | - | - |
| SAM2 3D (13) | Hiera | ✗ | ✗ | ✓/✗ | ✓/✓ | ✓/✗ | ✓/✓ | ✓ |
| SAM-Med3D (10) | 3D ViT | ✓ | (✓) | ✗/✗ | ✓/✓ | ✗/✗ | ✓/✗ | ✗ |
| SegVol (11) | 3D ViT | ✓ | ✓ | ✗/✗ | ✓/✓ | ✗/✗ | ✓/✓ | ✗ |
| MedicoSAM 3D (9) | SAM | ✓ | ✓ | ✓/✗ | ✓/✓ | ✓/✗ | ✓/✗ | ✗ |
| Vista3D (12) | SegResNet | ✓ | ✓ | ✗/✗ | ✓/✓ | ✗/✗ | ✓/✓ | ✗ |
| nnInteractive (4) | CNN | ✓ | ✓ | ✓/✓ | ✓/✓ | ✓/✗ | ✓/✓ | ✗ |
| Med-SAM2 (1) | SAM2 | ✓ | ✓ | ✓/✗ | ✗/✗ | ✗/✗ | ✓/✓ | ✓ |

**2D models evaluated slice-wise:** SAM Vit-B, SAM Vit-H, SAM Vit-L (5); SAM2.1 B+, SAM2.1 L, SAM2.1 S, SAM2.1 T for 2D image segmentation (13); Med-SAM (6), SAM-Med2D (7), ScribblePrompt-UNet and ScribblePrompt-SAM (8), MedicoSAM2D (9)

**3D models evaluated slice-wise *and* 3D models evaluated volumetric:** SAM2.1 B+, SAM2.1 L, SAM2.1 S, SAM2.1 T for 3D volume segmentation (13); Med-SAM2 (1), nnInteractive (4), MedicoSAM3D (9), SegVol (11), Vista3D (12).

## 2.2 Prompting strategies

For this study, we used non-iterative prompts, automatically extracted from the reference masks. The **2D prompting strategies** included bounding boxes, center points, or a combination of both, applied to up to five components. This choice was motivated by their effectiveness in 2D SAM-family models (14) and by the dataset characteristics, where most objects consist of up to five disconnected components. The **3D prompting strategies** extended the 2D approaches by incorporating the intial slice as a third dimension. All FMs (except SegVol (11)) rely on peudo-3D boxes represented by 2D coordinates and slice indices, which can be similarly applied to a 3D point. The chosen settings were bounding boxes, center points or their combination extracted from the largest component in a single initial slice, since this is the minimal feasible configuration compatible with all models (Table 1).

## 2.3 Dataset

As medical FMs are usually trained with publicly available datasets (15; 16; 17), to ensure a fair comparison across models, we had to resort to a private CT test dataset, approved by the local Medical Ethics Committee. To allow reproducibility, we included 10 samples of the TotalSegmentator test set (15) to our study. The final test dataset contains four skeletal regions: (D1) 10 bilateral shoulder CT scans with labels for scapula and humerus; (D2) 10 unilateral wrist CT scans with labels for capitate, lunate, radius, scaphoid, triquetrum, and ulna; (D3) 15 unilateral lower leg CT scans with labels for tibia and tibial implant; (D4) 10 unilateral hip CT scans (15) with the original labels hip and femur, and manually added labels for femur implant.
To reduce computational resources, we selected random slices as *initial slices* based on the following strategy: From D1 and D2, two slices per class were extracted; from D4, two slices per original class; and from D3, three slices per class. The top and bottom 10% of each volume were excluded to avoid slices with little relevant anatomy. To ensure spatial diversity, a minimum spacing between selected slices was enforced: $> 10$ slices for D1, D3, and D4, and $> 5$ slices for D2. In total, 370 slices were extracted (D1: 80, D2: 120, D3: 90, D4: 80).

## 2.4 Evaluation

**Segmentation Performance** was measured by Dice Similarity Coefficient (DSC), 95%-percentile Hausdorff distance (HD95), and Normalized Surface Dice (NSD) ($\tau = 1.5$ mm, based on largest

Table 2: Segmentation performance and model size for Pareto-optimal models in each category. Models highlighted in bold are Pareto-optimal and have the least model parameters.

| Model | Size (M) | DSC (↑) (%) | NSD (↑) (%) | HD95 (↓) (mm) | DSC (↑) (%) | NSD (↑) (%) | HD95 (↓) (mm) | DSC (↑) (%) | NSD (↑) (%) | HD95 (↓) (mm) |
|---|---|---|---|---|---|---|---|---|---|---|
| | | 2D Models | | | 3D Models evaluated slice-wise | | | 3D Models evaluated volumetric | | |
| *Prompt: Bounding box* | | | | | | | | | | |
| **Med-Sam2** | 39 | - | - | - | - | - | - | 79.56 ±11.1 | 80.25 ±10.5 | 13.49 ±11.1 |
| MedicoSAM2D | 94 | 90.74 ±7.7 | 97.36 ±3.6 | 0.76 ±0.9 | - | - | - | - | - | - |
| **Sam2.1 B+** | 81 | **90.60** ±8.1 | **97.84** ±3.5 | **0.82** ±1.0 | - | - | - | - | - | - |
| **nnInteractive** | 102 | - | - | - | **90.80** ±9.2 | **97.57** ±4.6 | **1.05** ±2.2 | - | - | - |
| *Prompt: Center point* | | | | | | | | | | |
| **Sam B** | 94 | **85.43** ±14.4 | **90.82** ±13.0 | **4.83** ±6.3 | - | - | - | - | - | - |
| **nnInteractive** | 102 | - | - | - | **83.47** ±12.9 | **90.92** ±9.6 | **2.16** ±2.8 | **69.40** ±11.2 | **68.23** ±12.0 | **30.98** ±9.4 |
| *Prompt: Combination of bounding box and center point* | | | | | | | | | | |
| MedicoSAM2D | 94 | 91.27 ±7.4 | 97.74 ±3.3 | 0.69 ±0.8 | - | - | - | - | - | - |
| **Sam2.1 B+** | 81 | 91.98 ±7.2 | 98.21 ±3.6 | 0.73 ±1.1 | - | - | - | **68.33** ±9.4 | **67.86** ±10.2 | **26.04** ±18.2 |
| Sam2.1 L | 224 | 90.90 ±6.9 | 98.36 ±3.2 | 0.69 ±1.0 | - | - | - | - | - | - |
| Sam2.1 S | 46 | 91.51 ±7.0 | 98.40 ±3.3 | 0.69 ±0.9 | - | - | - | - | - | - |
| **Sam2.1 T** | 39 | **91.83** ±6.9 | **98.38** ±3.2 | **0.71** ±1.0 | **89.27** ±8.6 | **96.95** ±5.5 | **1.95** ±3.4 | - | - | - |
| nnInteractive | 102 | - | - | - | 89.57 ±8.6 | 96.34 ±4.8 | 1.10 ±1.6 | 75.92 ±9.4 | 76.60 ±9.6 | 26.53 ±10.3 |

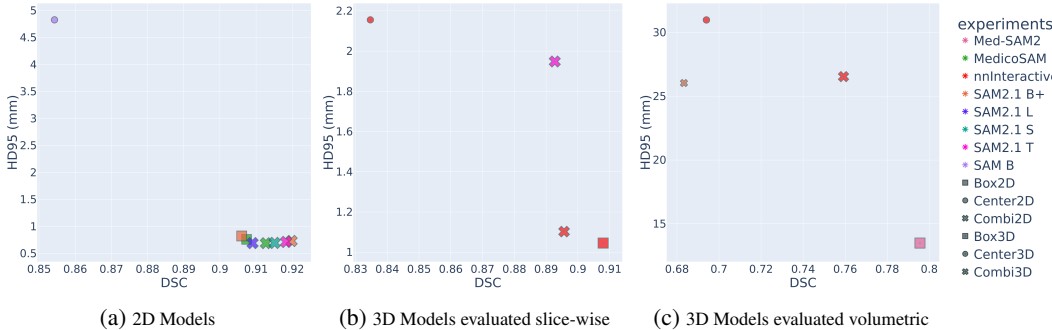

(a) 2D Models  (b) 3D Models evaluated slice-wise  (c) 3D Models evaluated volumetric

Figure 1: Segmentation performance of Pareto-front models (DSC vs. HD95 (mm)). Symbol size encodes NSD, mapped linearly to values between 2 (smallest) to 5 (largest).

spacing), following MetricsReloaded (18) and employing DisTorch (19). 2D and 3D models are evaluated on the selected slices (slice-wise), 3D models additionally on full volumes (volumetric). The **Pareto front** consists of all models that are not simultaneously outperformed across all criteria, i.e., no other model performs at least as well on every criteria and strictly better on at least one. For each of the three categories per prompt strategy, the best models in the Pareto sense are identified based on segmentation metrics (i.e., DSC, HD95, NSD). For multiple Pareto-optimal models, the model size (i.e., number of parameters) is also taken into consideration, jointly reflecting on performance and computational efficiency.

## 3 Results & Discussion

The Pareto-optimal models based on segmentation metrics for each category are illustrated in a DSC vs. HD95 (mm) scatterplot (Figure 1) and summarized in Table 2, with models that are Pareto-optimal and have the lowest model size highlighted in bold. The two main insights are: The bounding box prompt performs well in 2D and 3D, with improved results in 3D without combining with center points; Within the pool of models, medical dedicated FMs, such as MedicoSAM2D, Med-Sam2, and nnInteractive can perform on-par or outperform Sam and Sam2.1.

## 4 Conclusion

We identified the Pareto-optimal models based on segmentation metrics out of 11 promptable FMs using non-iterative 2D and 3D prompts. Although the *Pareto front* depends on the chosen criteria and different metrics may yield different selections, it avoids prioritizing a single metric. For future work, we are aiming to evaluate the prompt robustness of the selected models.

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
