# OpenReview forum: "Zero-shot evaluation of promptable foundation models for 3D CT segmentation"
_EurIPS.cc/2025/Workshop/MedEurIPS — EurIPS 2025 Workshop MedEurIPS Submission_

### Official Review · Reviewer_L6L8 · 2025-10-27
**Zero-shot evaluation of promptable foundation models for 3D CT segmentation**

**Rating:** 5
**Confidence:** 4

**Review:**

Summary: The paper evaluates and compares several prominent promptable medical and non-medical foundation models. The models are assessed using three metrics, and their performance trade-offs are analyzed through Pareto front identification.

Strengths:
- Foundation models are currently a key topic in the medical AI community, and systematic evaluation is essential for understanding their relative strengths and limitations.

Weaknesses:
- The analysis and insights presented in the paper are limited. The main value of such a study is providing a deeper understanding of the differences between models and explaining what contributes to their performance. However, this is not adequately addressed.
- The references and citations are incorrectly formatted and do not follow the standard citation style.

Overall Assessment:
The paper offers a comparative evaluation of multiple promptable foundation models but lacks sufficient depth in analysis and interpretation of results. A more thorough discussion of why certain models perform better and the implications of the findings would considerably strengthen the work.

---

### Official Review · Reviewer_5Gif · 2025-10-31
**Review comments**

**Rating:** 4
**Confidence:** 4

**Review:**

The paper evaluates 11 promptable foundation models for medical image segmentation, using bone segmentation as the task. The evaluation is significantly limited by the use of a small, private dataset (only 380 slices across four regions). Furthermore, the paper lacks in-depth analysis or justification for the observed model behaviors during interactive segmentation.

Despite testing 11 models, the study's findings offer limited practical guidance due to the constrained evaluation scope and lack of detailed analysis.

---

### Official Review · Reviewer_shG3 · 2025-10-31
**Review -  Zero-shot evaluation of promptable foundation models for 3D CT segmentation**

**Rating:** 6
**Confidence:** 3

**Review:**

The authors present a comparative study of currently existing promptable segmentation methods on the task of bone segmentation in 2D and 3D.

Strengths:
- The paper addresses a very important problem. With the large number of foundation models available, it is often difficult to assess which model to use for specific tasks.
- The evaluation includes a large number of different models that are well established in the literature.
- The goal of answering which model performs best for this specific task was achieved for this very specific datasets. This means that the paper has a nice take-home message.

Weaknesses:
- Showing DSC and NSD in the same plot feels a bit excessive. Since they are generally used for two different purposes (overlap vs. boundary focus), I recommend presenting them in two separate plots. If anything, make NSD clearer by not encoding it as size. Also, the dot colors seem inconsistent.
- Line 79 states that Figure 1 shows only the Pareto front for the models, but the scatterplot includes non-Pareto-optimal points (e.g., the pink one in the middle). It is also difficult to map the points to the experiments, as the legend does not match the points in the scatterplot.
- While the results are interesting for researchers working on bone segmentation, there is no indication of whether this work will be extended to other datasets. The paper also does not attempt to investigate underlying principles that could generalize beyond the scope of bone segmentation.

I believe that this paper will lead to a lot of great discussion and therfore recommend acceptance.

---

### Decision · Program_Chairs · 2025-10-31

**Decision:**

Reject

**Comment:**

The reviewers agree that the topic is timely and relevant, addressing the evaluation of promptable foundation models for medical image segmentation. However, they note that the study lacks depth in analysis, relies on a small private dataset, and offers limited generalizable insights.